# Determination of Multiclass Cyanotoxins in Blue-Green Algae (BGA) Dietary Supplements Using Hydrophilic Interaction Liquid Chromatography-Tandem Mass Spectrometry

**DOI:** 10.3390/toxins15020127

**Published:** 2023-02-04

**Authors:** María del Mar Aparicio-Muriana, Francisco J. Lara, Monsalud Del Olmo-Iruela, Ana M. García-Campaña

**Affiliations:** Department of Analytical Chemistry, Faculty of Sciences, University of Granada, Av. Fuente Nueva s/n, 18071 Granada, Spain

**Keywords:** blue-green algae, cyanotoxins, dietary supplements, HILIC-MS/MS, tandem-solid phase extraction

## Abstract

In recent years, the consumption of blue-green algae (BGA) dietary supplements is increasing because of their health benefits. However, cyanobacteria can produce cyanotoxins, which present serious health risks. In this work we propose hydrophilic interaction liquid chromatography coupled with tandem mass spectrometry (HILIC-MS/MS) to determine cyanotoxins in BGA dietary supplements. Target toxins, including microcystin-leucine-arginine (MC-LR) and microcystin-arginine-arginine (MC-RR), nodularin, anatoxin-a and three non-protein amino acids, β-N-methylamino-L-alanine (BMAA), 2,4-diaminobutyric acid (DAB) and N-(2-aminoethyl)glycine (AEG), were separated using a SeQuant ZIC-HILIC column. Cyanotoxin extraction was based on solid–liquid extraction (SLE) followed by a tandem-solid phase extraction (SPE) procedure using Strata-X and mixed-mode cation-exchange (MCX) cartridges. The method was validated for BGA dietary supplements obtaining quantification limits from 60 to 300 µg·kg^−1^. Nine different commercial supplements were analyzed, and DAB, AEG, and MCs were found in some samples, highlighting the relevance of monitoring these substances as precaution measures for the safe consumption of these products.

## 1. Introduction

Cyanobacteria, also known as blue-green algae (BGA), are a structural and morphologically diverse group of oxygenic photosynthetic prokaryotes, which is believed to be one of the oldest life forms on Earth. Cyanobacteria are well adapted to extreme environments such as drought, high ultraviolet radiation, low light, lack of nutrients and high salt concentrations. They live in a wide range of ecosystems such as lakes, ponds, rivers, marine environments, etc. Most cyanobacteria are an innocuous source of natural products with applications in the pharmaceutical, food, cosmetic, agricultural or fuel industry, however, there are at least 12 different species of cyanobacteria that have been proved to produce cyanotoxins [1], which are toxic secondary metabolites that can impact on ecosystem, animal and human health [2]. Numerous environmental factors such as temperature, light, pH and nutrients have been demonstrated to have a significant effect on cyanobacteria growth [3]. Moreover, human activity has led to an increase and hazardous spread of these algae blooms [3].

Humans can be exposed to cyanotoxins through a variety of routes, but the most frequent occurs through the consumption of contaminated water or foodstuff, especially fish, crops irrigated with contaminated water or algae-based dietary supplements [4]. At present, BGA production provides biomass for biofertilizers, animal feed, cosmetics, nutraceuticals, food, beverages and pigments in the food industry [5,6]. In fact, BGA-based dietary supplements have become well known around the world as a result of the increase in health awareness and disease prevention and also due to the demand for a natural alternative to pharmaceutical products [7]. Most of the BGA dietary supplements derive from two filamentous genera of cyanobacteria and one microalgae: *Spirulina* including *S. platensis* and *S. maxima*, *Aphanizomenon flos-aquae* and *Chlorella pyrenoidosa* [8]. *Spirulina* is usually produced under cultured conditions and in open tank systems. It is non-toxic as no direct production of toxins has been reported and it is considered a superfood: a great source of pigments, antioxidant compounds, vitamins, carbohydrates and proteins [9]. For this reason, and because it can be easily harvested and processed, food manufacturers are using *spirulina* as the preferred cyanobacterial strain to prepare healthy products and dietary supplements. *Aphanizomenon flos-aquae* is mainly harvested in open environment such as lakes, the most important being the Upper Klamath Lake in Oregon (USA). This cyanobacterium can produce toxins itself [10], such as anatoxins, saxitoxins, cylindrospermopsin and β-methylamino-L-alanine (BMAA). The green algae *Chlorella* is harvested in artificial ponds, thus contamination with other toxin producer organisms is not common. However, *Spirulina*, *Aphanizomenon* and *Chlorella* strains coexist in the environment with other cyanobacteria species such as *Microcystis*, which may cause possible contamination of the BGA supplements [11], and therefore safety assessments of BGA species are necessary before making a recommendation for consumption.

The studies and the risk management decision about BGA products are relatively recent. Most of the studies related to cyanotoxins are focused on the hepatotoxic microcystins (MCs) [12], which are produced by certain strains of some genera of cyanobacteria such as *Microcystis*, *Anabaena*, *Nostoc* and *Oscillatoria,* among others. They are specifically focused on a single congener out of the almost 250 MCs known, because most of the available toxicological data is based on microcystin-leucine-arginine (MC-LR). While legislation on the presence of MC-LR in waters for different purposes is clear [13], toxicological analyses are not compulsory for dietary supplements. Considering the information provided by the toxicological studies, tolerable daily intake (TDI) and guidance values have been proposed by some institutions to prevent cyanotoxin contamination. For instance, the Oregon Health Division and the Oregon Department of Agriculture established a regulatory limit of 1 μg·g^−1^ for MCs in BGA-containing products [14]. The World Health Organization (WHO) established a TDI of 0.04 µg·kg^−1^ body weight (bw) for chronic exposure to MC-LR and recommends a safe limit of 1 µg·L^−1^ in drinking water [15]. According to these guidance levels, infants (5 kg), children (20 kg) and adults (60 kg) could tolerate a maximum exposure of 0.2, 0.8 and 2.4 μg MC-LR·day^−1^, respectively. In the absence of information on the other variant, concentration MC-LR equivalents are usually used as default value for the total concentration of all MC congeners. Regarding the rest of cyanotoxins, there is not an established TDI due to the scarcity of information available, thus no other provisional guidance values for cyanotoxins have been set by the WHO.

In view of the potential poisoning problem associated with the consumption of BGA dietary supplements and the fact that the algae *Aphanizomenon flos-aquae* produces toxins itself, a better understanding of the risks and toxic effects as well as the monitoring of these substances in BGA dietary supplements is desired. To achieve a quantitative determination of cyanotoxins, it is necessary to consider their chemical properties, from the highly polar non-protein amino acids (NPAs), such as BMAA, 2,4-diaminobutyric acid (DAB), and N-(2-aminoethyl)glycine (AEG), to the more lipophilic MCs, which complicates the extraction procedure and adds difficulty when a combined simultaneous extraction and analysis is desired. In this sense, few studies have been conducted on cyanobacteria and dietary supplements derived from BGA. Most of the proposed analytical methods are focused on a single family of compounds. For instance, NPAs, such as BMAA and their isomers DAB and AEG, have been determined through two ways: (1) without pre-analysis derivatization using hydrophilic interaction chromatography coupled to tandem mass spectrometry (HILIC-MS/MS) and (2) derivatized with different reagents such as 9-fluorenylmethyl chloroformate (FMOC), propyl chloroformate (PCF) and especially 6-aminoquinolyl-N-hydroxysuccinimidyl carbamate (AQC). In that case, the determination is performed by reverse phase liquid chromatography (RPLC) coupled with different detection systems such as MS/MS, ultraviolet (UV) and fluorescence (FL). These contributions using RPLC have been discussed and summarized in a review [16]. MCs from the cyclic peptide family have been determined using RPLC coupled to MS/MS [11,12,17,18,19,20] or to high-resolution mass spectrometry (HRMS) [21,22,23]. Alkaloids such as cylindrospermopsin, and anatoxin-a (ANA) and its metabolites have also been studied separately using mainly RPLC-FL, although UV and mass spectrometry (MS) detectors have also been employed [24,25,26,27,28]. Regarding multi-toxin analysis, some studies cover the presence of diverse congeners and cyanotoxin families in dietary supplements, especially MCs, cylindrospermopsin, saxitoxin, anatoxins and some of them also included BMAA. These methods apply different extraction procedures, depending on the family and also the chromatographic method is developed separately for lipophilic and hydrophilic toxins [8,29,30,31]. In those cases, sample treatment approaches are generally based on solid–liquid extraction (SLE) using aqueous mixtures of about 75% methanol, acidified methanol or acidified water, depending on whether cyclic peptides, alkaloids or NPAs are extracted. Then, the extracts are frequently cleaned by solid-phase extraction (SPE) using hydrophobic cartridges for lipophilic cyanotoxins such as MCs and nodularin (NOD) and cation exchange cartridges for the hydrophilic cyanotoxins such as BMAA [29]. More recently, some studies encompassed the combined extraction and determination of cyanotoxins belonging to two cyanotoxin families, such as cyclic peptides and alkaloids in BGA dietary supplements [32,33]. They employ a similar extraction protocol with 75% of aqueous methanol as extractant solvent without further purification or clean-up step and cyanotoxins are determined by RPLC-MS/MS. However, to the best of our knowledge, the simultaneous extraction and determination of the three NPA isomers BMAA, DAB and AEG, the cyclic peptides MC-LR, microcystin-arginine-arginine (MC-RR) and NOD and the alkaloid ANA from BGA dietary supplements has not been carried out to date. In addition to the technical challenges associated with an accurate extraction, detection, quantification and interpretation of the results of multi-cyanotoxin analysis in BGA dietary supplements, some of them such as MCs and BMAA can exist in their free form or bound to proteins, in which case it is important to use a sample treatment fitted for the intended purpose, and it is necessary to indicate whether the free, bounded or total fraction has been studied.

In a previous work, we established an UHPLC-MS/MS method for the simultaneous determination of different classes of cyanotoxins in reservoir waters. A tandem-SPE procedure was employed to extract and preconcentrate the analytes [34]. Based on the progress achieved in such work, the main goal of this study is to propose an effective analytical method for comprehensive determination of the free fraction of seven cyanotoxins, belonging to the cyclic peptide family (MC-LR, MC-RR and NOD) [35], alkaloid family (ANA) [36], and NPAs (BMAA, DAB and AEG) [37], from BGA and microalgae-derived dietary supplements. Among all the existing cyanotoxins, the above mentioned have been selected to represent each of the existing families. For instance, MCs are a family of almost 250 structurally similar hepatotoxins, and the most important ones have been presented here. One of them is MC-LR, whose choice is essential since it is the most extensively studied and the only microcystin that is regulated. The second choice is MC-RR, because according to the literature, it is the second most common cyanotoxin in the environment [38]. A SLE has been optimized to extract the diverse analytes from the complex matrix into a solvent that allows subsequent cleaning by means of SPE. This work provides a new procedure for the multiclass determination of cyanotoxins of different polarities in this kind of BGA dietary supplements. Likewise, it is intended to evaluate the feasibility, challenges and pitfalls that arise in the field of multi-toxin analysis when they are carried out in such complex matrixes.

## 2. Results and Discussion

### 2.1. Optimization of Chromatographic and MS/MS Conditions

The chromatographic separation and MS/MS detection conditions employed were based on our previous work for the analysis of these toxins in reservoir water samples [34]. At this initial study, different stationary phases were tested; reversed phase (RP) columns (Zorbax RRHD Eclipse Plus C18, Luna Omega C18 Polar and Kinetex Biphenyl) and HILIC columns (Kinetex HILIC and SeQuant ZIC-HILIC). RP columns gave either no significant or slight retention of the studied compounds resulting in broad peaks. Among the two tested HILIC columns, SeQuant ZIC-HILIC, which has a sulfobetaine type zwitterionic stationary phase covalently attached to silica particles, showed the best results in terms of resolution, although the isomers were not eluted under the isocratic mode. A minor gradient composition adjustment was considered in relation to this previous study, due to the complexity of the *spirulina* matrix. Regarding the MS/MS detection, the precursor and product ions of individual analytes were those previously identified in our laboratory by tuning after direct infusion of individual standard solutions of 1 mg·L^−1^ in 0.1% aqueous formic acid:methanol (50:50, *v/v*) into electrospray ionization tandem mass spectrometry (ESI-MS/MS) system. All the target cyanotoxins were determined using the protonated form ([M + H]^+^) as a precursor ion except for MC-RR, which tends to ionize as the diprotonated molecular ion ([M + 2H]^2+^). The fragment ion with the highest intensity was used for quantification (Q_ion_), whereas the second one was acquired for identification (I_ion_). However, it is worth highlighting the fact that in the case of ANA, the most abundant multiple reaction monitoring (MRM) transition in standard solution, corresponding to *m/z* 166.1 > 149.0 (Figure 1A), could not eventually be used as Q_ion_ for the analysis of BGA samples because an interference peak with this same transition appeared at the same retention time than ANA (4.8 min). This peak did not correspond to ANA because it did not present any of the other characteristic transitions of ANA (Figure 1B). Thus, the fragment ion with the second highest intensity was selected as Q_ion_ (166.1 > 131.1), and the one with the third highest intensity was used as I_ion_ (166.1 > 43.1) for ANA. Furthermore, an intense peak with the same precursor and product ions than ANA but with different ion ratios appeared at 4.0 min just before ANA peak (Figure 1C). According to the literature, this peak corresponds to the essential amino acid Phenylalanine (Phe), which is an isobaric compound of ANA and presents a similar fragmentation pattern and liquid chromatography (LC) retention [39]. In fact, misidentification of Phe as ANA has been reported in forensic investigation, especially in the presence of naturally occurring Phe. The assumption that this peak corresponds to Phe is also supported by the fact that *spirulina* contains a moderately high amount of both essential and non-essential amino acids, being the content of Phe around 28 mg·g^−1^ [40].

### 2.2. Multi-Toxin Extraction Procedure

Cyanotoxin extraction from BGA supplements is influenced by a wide range of features such as physico-chemical properties and nature of the matrix. Because of this, most of the developed analytical methods do not include the extraction of a multitude of congeners, and different approaches are usually employed for each family of toxins including differences in the extraction procedure, but also in the separation method.

To extract the target cyanotoxins showing different polarities from the BGA supplements, a SLE was proposed followed by an isolation and clean-up step based on a tandem-SPE. Thus, it is possible to remove the remaining matrix, minimizing interferences, and to achieve a preconcentration of the analytes increasing sensitivity, prior their injection into the HILIC-MS/MS system. For the sample treatment optimization, a dehydrated organic *spirulina* sample free of cyanotoxins was spiked with a mixture of the target toxins at concentration levels of 200 μg·kg^−1^ for MC-LR, MC-RR, NOD and ANA and 1500 μg·kg^−1^ for the isomers BMAA, DAB and AEG, depending on the analyte sensitivity. Optimum extraction conditions were selected for achieving the highest recoveries, which were calculated by comparing the peak area of the analytes detected from a BGA sample spiked before and after the extraction procedure. Several parameters were assessed with the aim of improving the efficiency of the SLE procedure, being the more significant the extraction solvents. BMAA and its isomers can be associated in the cell substrate mainly as free or protein-bound forms. For the purpose of this paper, only the free cyanotoxin fractions were considered.

#### 2.2.1. Study of Solvents for Solid–liquid Extraction

Free BMAA and its isomers can be extracted from lyophilized samples or raw powder with a suitable solvent, typically 0.1–0.3 M trichloroacetic acid (TCA), although other solvents have also been used with this purpose, such as hydrochloric acid (HCl), acetonitrile (MeCN), methanol (MeOH) and acidified water with citric acid [16]. Regarding MCs, polar extractants such as mixtures of water with a polar or mid-polar organic solvent showed high recoveries [41], and when cyanobacteria were treated with MeOH and HCl or acetic acid, good yields were obtained for anatoxins. Firstly, mixtures of MeOH, MeCN and ethanol (EtOH) with water (80:20, *v/v*) were tested as extraction solvents in BGA samples. This ratio was chosen according to the literature, where around 75–90% methanolic-water mixtures are usually employed to extract MCs and NOD with optimal recoveries from dietary supplements [8,11,20,30]. With this purpose, 50 mg of organic spirulina spiked with a mixture of cyanotoxins were extracted with 4 mL of the extraction mixtures, stirred 10 min and placed in an ultrasonic bath for 15 min. After centrifuging at 9000 rpm and 4 °C during 10 min, the supernatant was transferred to a 50 mL centrifuge tube and made up to 32 mL with deionized water to achieve a 10% organic solvent (MeOH) in the extract for loading onto tandem-SPE. The aqueous extract was submitted to the tandem-SPE procedure previously described in [34]. Among all the tested solvents, 80% MeOH showed the best results with recoveries above 70% for MCs and NOD, and about 60% for ANA (Figure 2A). However, a deficient extraction of NPAs was obtained, with recoveries below 50%. No improvement in recoveries was obtained when two extractions were made halving the extraction volume, so one extraction with 4 mL of 80% aqueous MeOH was maintained. Afterwards, with the aim of improving extraction efficiency of the NPAs, an acidic aqueous extraction was evaluated before the methanolic one as well as increasing the sonication time. With this objective, 4 mL of trichloroacetic acid (TCA) 0.1 M and 20 min of sonication bath were employed. After centrifuging, the supernatant was transferred to a 50 mL centrifuge tube and stored. Then, 4 mL of 80% MeOH were added to the remaining pellet to repeat the extraction. The collected supernatants were combined and made up to 32 mL with water. The aqueous extract was submitted to the tandem-SPE procedure previously described and injected into the HILIC-MS/MS system. As expected, the addition of a first 0.1M TCA extraction before the methanolic one significantly increased the recoveries of the NPA isomers up to around 70% while recovery value for the MCs and NOD remained unchanging at around 80% (Figure 2B). However, it is remarkable that under these conditions, the peak corresponding to ANA, which should have appeared at 4.8 min, was missing, and only the intense peak of Phe was observed between 3.0 and 3.7 min (Figure 3A). It was supposed that the remaining TCA present in the injected sample negatively affected the interaction of ANA with the column, reducing its retention time and causing its coelution with the big peak of Phe, precluding its determination. In the light of the obtained results, it seems clear that an aqueous acidic extraction enhances the extraction of NPAs, so the next step was to compare the extraction performance of a variety of acidic solvents. With this purpose, 4 mL of 0.1 M TCA, 0.1 M oxalic acid (OA) and 5% formic acid (FA) were evaluated as the first extraction step in the SLE procedure while keeping constant the rest of the procedure. Similar recoveries were obtained when using TCA and OA, being over 70% for NPAs and about 80% for MCs and NOD while ANA peak remained missing when both extraction solvents were employed. On the contrary, when 4 mL of 5% FA was employed, the ANA peak was observed at 4.8 min, just after the large Phe peak (Figure 3B), with an acceptable recovery of 60% (Figure 2C).

However, recoveries for NPAs declined below 60%, and the reproducibility worsened, so these problems were explored. Moreover, the final reconstituted extract after the SPE procedure tended to show a two-phase separation. This may be attributed to the remaining salts from the BGA supplements in the dried extract, which caused a sort of salting-out effect when re-dissolving it in the 60% aqueous MeCN with 0.1% FA before the injection.

#### 2.2.2. SPE Washing Step

In this respect, the addition of a washing step after sample loading in the tandem-SPE procedure was critical for matrix removal and for the complete re-dissolution of the dried extract in the injection solvent. The washing solvent optimization study was performed for each cartridge separately. On the one hand, for the Strata-X cartridge, 2 mL of different percentages of MeOH in water (from 0 to 50%) were tested to determine the maximum percentage that can be tolerated without any significant analyte loss. A 30% aqueous MeOH yielded the best results, as a higher amount of MeOH resulted in a decrease in MC-LR recoveries (Appendix A). On the other hand, the washing solvents evaluated for mixed-mode cation-exchange (MCX) cartridge were 2 mL of different solvents such as: 100% MeOH, 30% MeOH, 2% FA in MeOH, and a two-step washing procedure with 2% aqueous FA followed by 100% MeOH, which is commonly used according to the protocol specified by the manufacturer for MCX cartridge. As expected, when analytes are strongly held by an ionic attraction, a 100% of MeOH proved to be a powerful washing step to remove matrix interferences. According to the data obtained, a 30% aqueous MeOH was selected as the optimum washing solvent for MCX cartridge because similar recoveries to those using 100% MeOH were obtained for all analytes (Appendix A in Supplementary Materials) and in this way the dual-cartridge system could be washed in a single step. The washing volume passing through the cartridges was evaluated between 1 and 3 mL. A volume of 2 mL of 30% aqueous MeOH was selected as satisfactory matrix removal, which was attained while limiting analyte breakthrough. With this washing step, a single-phase final extract was achieved, overcoming the drawback of the two-phase separation in the final re-dissolved extract, which involved injection irreproducibility.

#### 2.2.3. SPE Load Volume

After that, considering that a percentage of 30% of MeOH in the washing solvent was tolerated, the increase in the percentage of MeOH in the SPE load solvent was modified in order to speed up the loading step. Therefore, when supernatants from SLE extraction were combined, they were filled with deionized water up to only 16 mL to reach a 20% MeOH in the final extract to be loaded. Finally, the amount of BGA sample to be treated was studied between 50 and 100 mg, being 75 mg the optimum value as higher amounts of sample decreased the MCs and NOD recoveries. The final procedure is described in Section 4.4 and Section 4.5.

A chromatogram of a BGA dietary supplement sample spiked with the studied analytes submitted to the optimized sample treatment and analyzed by the proposed SLE-SPE-HILIC-MS/MS method is shown in Supplementary Materials (Appendix A).

### 2.3. Validation of the Method in Spirulina Sample

The proposed method was validated in terms of linearity, sensitivity, limit of detection (LOD) and limit of quantification (LOQ), extraction efficiency, precision (i.e., repeatability and intermediate precision) and matrix effect, using organic *spirulina* purchased from a local retail store. To assess specificity, blank samples were analyzed, and no interferences of endogenous substances were observed comigrating at the retention times of the analytes of interest. Three concentration levels (L1, L2 and L3) were evaluated for precision, recovery, and matrix effect studies, which corresponded with the first (LOQ values), third and fourth concentration levels of the calibration curve for all analytes.

#### 2.3.1. Calibration Curves and Analytical Performance Characteristics

Procedural calibration curves were established in BGA dietary supplement samples. In this regard, 0.5 g of blank sample were spiked at the beginning of the extraction procedure with the desired concentration of the target cyanotoxins, left to stand 15 min, and then they were submitted to the optimized SLE-tandem-SPE-HILIC-MS/MS method previously described. Five concentration levels were evaluated, with the lowest level corresponding to the LOQ, ranging from 60 to 500 μg·kg^−1^ for MC-LR, from 50 to 500 μg·kg^−1^ for NOD and MC-RR, from 150 to 750 μg·kg^−1^ for ANA, from 300 to 2500 μg·kg^−1^ for BMAA and DAB and from 150 to 2500 μg·kg^−1^ for AEG. Three samples for each concentration level were processed and injected in triplicate (n = 9) following the optimized extraction procedure.

Peak area was considered as response signal, being linearly dependent on the analyte concentration in the sample. In all cases, satisfactory linearity was achieved over the working range (with determination coefficients R^2^ > 0.9817). Statistical parameters calculated by least-square regression and the performance characteristics of the method can be seen in Table 1. LODs were calculated as the lowest concentration for which the peak height of the Q_ion_ was, at least, three times the signal-to-noise ratio while LOQs were calculated as the lowest concentration with a signal-to-noise ratio of at least 10, which fulfilled method performance acceptability criteria of precision and extraction efficiency. For some analytes, such as the amino acids, the LOQ values obtained were higher than those provided by other studies. This is due to the complexity of the approach in terms of sample treatment, as there are different toxins with different properties that limit the simultaneous extraction to some extent. In any case, the obtained LOQs were in the low μg·kg^−1^ levels, which allows the detection and quantitation in real samples.

#### 2.3.2. Matrix Effect

The matrix effect (ME) was assessed at three concentration levels (L1, L2 and L3) by comparing the analyte response from a blank extract of BGA dietary supplement sample spiked with the analytes after sample treatment and just before injection and the analyte response from a solvent standard solution spiked with the analytes after sample treatment and just before injection, both at the same concentrations. This procedure eliminates the contribution of reagents and solvents to the matrix effect. ME was calculated over three samples for each concentration level and analyzed in triplicate (n = 9) according to the following Equation (1).
(1)$$ME\left(\%\right)=\frac{\left(Signal\;of\;analyte\;in\;sample\;extract\;spiked\;after\;treatment\right)}{\left(Signal\;of\;analyte\;in\;standard\;solution\;spiked\;after\;treatment\right)}\times 100$$

From this equation, a ME equal to 100% indicates the absence of a matrix effect, whereas a value greater or lower than 100% implies signal enhancement or ion suppression, respectively. Ion suppression was observed for almost all analytes to a greater or lesser extent, being the isomers BMAA and DAB the most affected, with ME values down to 9.1 and 21.5%, respectively, as it is shown in Table 1. Furthermore, it was noteworthy the fact that the ionization of MC-LR is the most matrix-dependent, as it exhibited a signal enhancement over 350%, even though in the blank extracts, no peak was observed at its retention time. MC-LR is the first analyte that elutes from the HILIC column as it is barely retained. Thus, it was hypothesized that endogenous matrix components poorly retained in the column elute along with MC-LR modifying in some way the ionization state of this analyte. Previous work has also shown a signal enhancement of MC-LR when analyzing dietary supplements [32], but not at such high levels as we obtained. Enhancement of ionization has the potential to be used in a favorable manner, improving sensitivity. With this purpose, further studies would be needed in this area to help understand the behavior of certain compounds in the presence of some additives. However, this limitation was compensated by performing a procedural calibration curve for quantification purposes, which considers the correction of bias introduced from the matrix effect and from losses in sample treatment.

#### 2.3.3. Precision

The precision of the method was assayed in terms of repeatability (intra-day precision) and intermediate precision (inter-day precision) by the application of the whole procedure to organic *spirulina* samples spiked at three different concentration levels and the comparison of the peak area values obtained. Repeatability was assessed over three samples for each concentration level and analyzed in triplicate (n = 9) on the same day under the same conditions. Similarly, intermediate precision was evaluated during five consecutive days by analyzing one sample per day in triplicate (n = 15). Results, expressed as relative standard deviation (% RSD) of peak areas, are shown in Table 1. Acceptable precision (RSD < 20%) was obtained with RSD values lower than 19.6% in all cases, except for BMAA, which reported the highest RSD values, reaching 25.1% at the lowest concentration level in the inter-day precision.

#### 2.3.4. Recovery Assays

Recovery experiments were carried out at the same L1, L2 and L3 concentration levels, by analyzing three real samples for each concentration level in triplicate (n = 9). Recovery at each concentration level was estimated by comparing the analyte concentrations obtained from samples spiked before and after the whole sample treatment (SLE-SPE procedure) using peak area as analytical signals. Results showed satisfactory recoveries ranging between 64.2% and 102.9% with an associated RSD < 19.2% for all analytes (Table 2).

### 2.4. Determination of Cyanotoxins in BGA Dietary Supplements

The SLE-tandem-SPE-HILIC-MS/MS method developed was satisfactorily applied to determine the natural occurrence of the target toxins in the BGA dietary supplements described in Section 4.3. For each sample, three replicates were analyzed in triplicate (n = 9). To the best of our knowledge this is the first time that this simultaneous analysis of multiclass cyanotoxins is carried out and applied in real samples. Eight of nine samples were found positive for at least one cyanotoxin (Table 3).

Among all the target cyanotoxins, DAB, AEG, MC-LR and MC-RR were present in some of the BGA supplements. DAB was the most frequent one, being found on seven out of nine samples at concentrations ranging from 331 to 2480 µg·kg^−1^. Three of these samples exceeded the concentration value of 1 mg·kg^−1^. AEG was detected in two samples but was only quantified in one of them (194 µg·kg^−1^). In general, the concentrations present in the samples did not exceed from 2.5 mg·kg^−1^. However, MC-LR and MC-RR were present in one sample of *Aphanizomenon flos-aquae* at levels higher than 5 mg·kg^−1^ (Figure 4), which is similar to that has been found in previous works [11,17].

In the present work, based on the daily dose recommended by each manufacturer for each BGA supplement, 2.4 g was chosen as the average daily ration. Considering this daily dose value and the TDI for the only regulated cyanotoxin, the MC-LR (2.4 µg per day), a value of 1 mg·kg^−1^ was obtained as maximum tolerable concentration of MC-LR in BGA supplements. This does not entail that MC-LR is the only toxic or that the rest of cyanotoxins are less harmful. It merely reflects the lack of toxicological information on these compounds. Due to the lack of data and according to precautionary principle, the same maximum tolerable value of 1 mg·kg^−1^ has been taken as a reference for the rest of cyanotoxins to conclude the exposure risk in BGA, even though they do not have a fixed TDI. Under this premise, results showed that three samples exceeded this value for DAB, reaching concentrations up to 2.48 µg·g^−1^. Furthermore, the high concentrations, up to 5 µg·g^−1^ for MCs, found in one sample of *Aphanizomenon flos-aquae* far outstripped the considered maximum tolerable value of 1 µg·g^−1^. These results are in line with those previously observed in other studies, where MCs were not present in *spirulina* samples, but they were repeatedly detected in *Aphanizomenon flos-aquae* (also known as Klamath) dietary supplements [8,11,18,30,42,43].

## 3. Conclusions

In this work, a multi-toxin method has been developed for the first time to determine simultaneously cyanotoxins from several common families with different chemical characteristics in BGA dietary supplements. For this purpose, a SLE procedure followed by a tandem-SPE were applied as sample treatment, using HILIC coupled to MS/MS as analytical technique. From a critical point of view, the main challenge posed in this work has been to achieve for the first time the analysis of multiclass cyanotoxins in such a complicated matrix in a single run, performing a single extraction of the analytes belonging to different chemical families and being able to quantify concentrations of each toxin at extremely low levels. It was possible to validate the method in *spirulina*-based dietary supplements, showing its usefulness for these kind of samples. Positive samples found for MCs and NPAs set a point of attention toward the need of a satisfactory quality control in the production of these dietary supplements and envisaged the importance of international legislation to ensure product safety and to protect consumer health from cyanotoxin contamination.

## 4. Materials and Methods

### 4.1. Reagents and Materials

Commercial standards for microcystin-leucine-arginine (MC-LR) and microcystin-arginine-arginine (MC-RR) (≥99% purity), nodularin (NOD) and anatoxin-a (ANA) (≥98% purity) were supplied by Enzo Life Sciences, Inc. (Lausen, Switzerland). Isomers β-N-methylamino-L-alanine hydrochloride (BMAA) (≥97% purity), 2-4-diaminobutyric acid dihydrochloride (DAB) (≥95% purity) and N-(2-aminoethyl)glycine (AEG) (≥98% purity) were supplied by Sigma Aldrich (Darmstadt, Germany). Appendix A, in Supplementary Materials, shows the structure and physico-chemical properties of the target cyanotoxins.

Stock standard solutions were prepared by adding 1 mL of the desired solvent directly into the vial of the toxin supplied by the manufacturer and gently swirling the vial to dissolve the analyte. The obtained solutions were: 50 μg·mL^−1^ MC-LR in methanol, 50 μg·mL^−1^ MC-RR in 80% aqueous MeOH, 50 μg·mL^−1^ NOD in 50% aqueous MeOH and 1000 μg·mL^−1^ ANA in water. Stock solutions of 1000 μg·mL^−1^ for the three standard isomer molecules (BMAA, DAB and AEG) were prepared by dissolving the desired amount of analyte in water. All of them were stored in the dark at −20 °C. Intermediate standard solutions of each compound at 2.5 μg·mL^−1^ were prepared by dilution of the stock solutions with the corresponding solvent for each toxin. Working standard multi-toxin solutions containing the seven cyanotoxins were prepared when needed in 50% aqueous MeOH at the desired concentrations, depending on the experiment. These solutions were stored at 4 °C and equilibrated to room temperature before use.

HPLC-grade acetonitrile (MeCN) and methanol (MeOH) were purchased from VWR (Radnor, PA, USA), ammonia solution (NH_3_·H_2_O) (30% assay) was purchased from Merck (Darmstadt, Germany), trichloroacetic acid was purchased from Panreac (Barcelona, Spain) and formic acid (FA) was purchased from Sigma-Aldrich (St. Louis, MO, USA). Ultra-pure water (18.2 MΩ·cm resistivity) was obtained from an ultrapure water purification system (Milli-Q plus system, Millipore, Bedford, MA, USA).

Oasis MCX cartridges (150 mg) from Waters (Milford, MA, USA) and Strata-X cartridges (200 mg) from Phenomenex (Torrance, CA, USA) were used for the cyanotoxin extraction procedure. SPE tube adapters from Supelco Inc. (Bellefonte, PA, USA) were employed. CLARIFY Polytetrafluoroethylene (PTFE) hydrophilic syringe filter (0.2 μm × 13 mm) were supplied by Phenomenex (Torrance, CA, USA).

### 4.2. Instrumentation

Chromatographic separation was performed on an Agilent 1290 Infinity II System (Agilent Technologies; Waldbronn, Germany) equipped with a quaternary pump, a degasser, an autosampler (with 20 μL injection loop) and a column thermostat. Separation was achieved using a SeQuant ZIC-HILIC column (2.1 × 250 mm, 3.5 μm diameter, EMD Millipore; Billerica, MA, USA). The HPLC system was coupled to an API 3200 triple quadrupole (QqQ) mass spectrometer (SCIEX; Darmstadt, Germany) equipped with a Turbo V electrospray ionization source. Instrumental data were collected by the Analyst^®^ Software (version 1.5) using the Scheduled MRMTM Algorithm (SCIEX).

For sample treatment, an analytical balance with 0.001 g resolution (Sartorius; Goettingen, Germany), a vortex-2 Genie (Scientific Industries; Bohemia, NY, USA), an Universal 320R centrifuge (Hettich Zentrifugen; Tuttlingen, Germany), a nitrogen dryer EVA-EC System (VLM GmbH; Bielefeld, Germany), a high speed solid crusher (Model 250A from Hukoer, China), an ultrasonic bath USC-300 model (VWR; Radnor, PA, USA), and a Visiprep solid-phase extraction unit from Supelco (Bellefonte, PA, USA) were used.

### 4.3. BGA Dietary Supplements

BGA dietary supplements were composed by the BGA *Spirulina* and *Aphanizomenon flos-aquae*, the microalgae *Chlorella*, and the brown algae *Fucus*, in different ratios or in pure form. They were obtained from several brands and different sources. They were sold as tablets, capsules and powder and were purchased from local retail stores and via the internet. Detailed information on all samples with their form, suppliers and BGA content are listed in Supplementary Materials Appendix A. All the samples were analyzed before their expiration date.

### 4.4. Analyte Extraction

Each dietary supplement was ground individually with a grinder to obtain a fine powder, and a representative subsample was taken for extraction. BGA products (75 mg) were weighed into a 15-mL conical centrifuge tube. Then, 4 mL of the first extraction solvent, which consists of 5% aqueous FA, were added to the sample and it was mixed by vortex for 1 min and placed in an ultrasonic bath for 20 min. Afterwards, the sample was centrifuged for 10 min at 9000 rpm and 4 °C and the supernatant was transferred to a 50 mL centrifuge tube. The residual pellet was reextracted with 4 mL of 80% aqueous MeOH. The mixture was shaken thoroughly by vortex for 1 min followed by sonication for 20 min in ultrasonic bath. After centrifuging at 9000 rpm and 4 °C for 10 min, the supernatant was collected and transferred to the 50 mL centrifuge tube containing the previous supernatant to combine both. The mixture was made up to a final volume of 16 mL with deionized water to obtain a 20% of MeOH in the solution. The crude extract is then purified on a combined “in-series” SPE system.

### 4.5. SPE Procedure

For the SPE procedure a previously reported method for the determination of cyanotoxins in reservoir water samples was employed with some modifications [34]. The SPE procedure was carried out using an assembly of two cartridges, a Strata-X (200 mg, 6 mL) and Oasis^®^ MCX (150 mg, 6 mL), connected in series. The two cartridges were conditioned and activated separately with 3 mL of MeOH followed by 3 mL of deionized water. Then, the cartridges were connected in series, being Strata-X cartridge at the top followed by the MCX cartridge, and the 16 mL extraction solution was loaded at a flow rate of 1 mL·min^−1^. Thereafter, the cartridge assembly was washed with 2 mL of 30% aqueous MeOH and dried under vacuum for 2 min. Before the elution step, the order of the cartridges was reversed, thus the MCX cartridge was configured first followed by the Strata-X cartridge. Elution was carried out using 5 mL of 10% NH_3_·H_2_O in MeOH. The eluate was evaporated to dryness under a gentle stream of nitrogen in a heating block (30 ˚C) and the residue was re-dissolved with 250 µL of 60% MeCN with 0.1% of FA. The final extract was filtered through a CLARIFY-PTFE hydrophilic filter (0.2 μm × 13 mm), transferred to a 0.3 mL glass insert and analyzed by the HILIC-MS/MS method.

### 4.6. HILIC Separation

Chromatographic separation was performed in a SeQuant ZIC-HILIC (2.1 × 250 mm, 3.5 μm particle size) column using a mobile phase consisted of water as eluent A and MeCN as eluent B, both containing 0.3% (*v/v*) of FA as volatile acid, at a flow rate of 0.2 mL·min^−1^. The gradient program was established as follows (expressed as a percentage (*v/v*) of eluent B in mobile phase): 0 min, 60%; 3 min, 60%; 6 min, 40%; 15 min, 40%. At 15 min, mobile phase composition went back to initial conditions in 1 min, and it was maintained for 20 min in order to guarantee column equilibration and reproducibility between runs. Column temperature was set at 55 °C and 20 μL (full loop injection) was selected as injection volume.

### 4.7. MS/MS Parameters

The determination of the target cyanotoxins was carried out using an electrospray ionization tandem mass spectrometry (ESI-MS/MS) system operated under positive mode in scheduled MRM conditions. The source parameters were set as follows: source temperature (TEM), 550 °C; ion spray (IS) voltage, 5500 V; nebulizing and drying gases (GS1 and GS2, nitrogen), 50 psi (344.7 kPa), curtain gas (CUR, nitrogen), 25 psi (172.4 kPa) and collision gas (CAD, nitrogen) was set to 10 psi (69.0 kPa). Main MRM parameters, including declustering potential (DP), entrance potential (EP), collision entrance potential (CEP), collision energy (CE) and collision cell exit potential (CXP), as well as precursor ions, product ions (used for quantification and identification purposes) and retention time are given in Table 4.

## Figures and Tables

**Figure 1 toxins-15-00127-f001:**
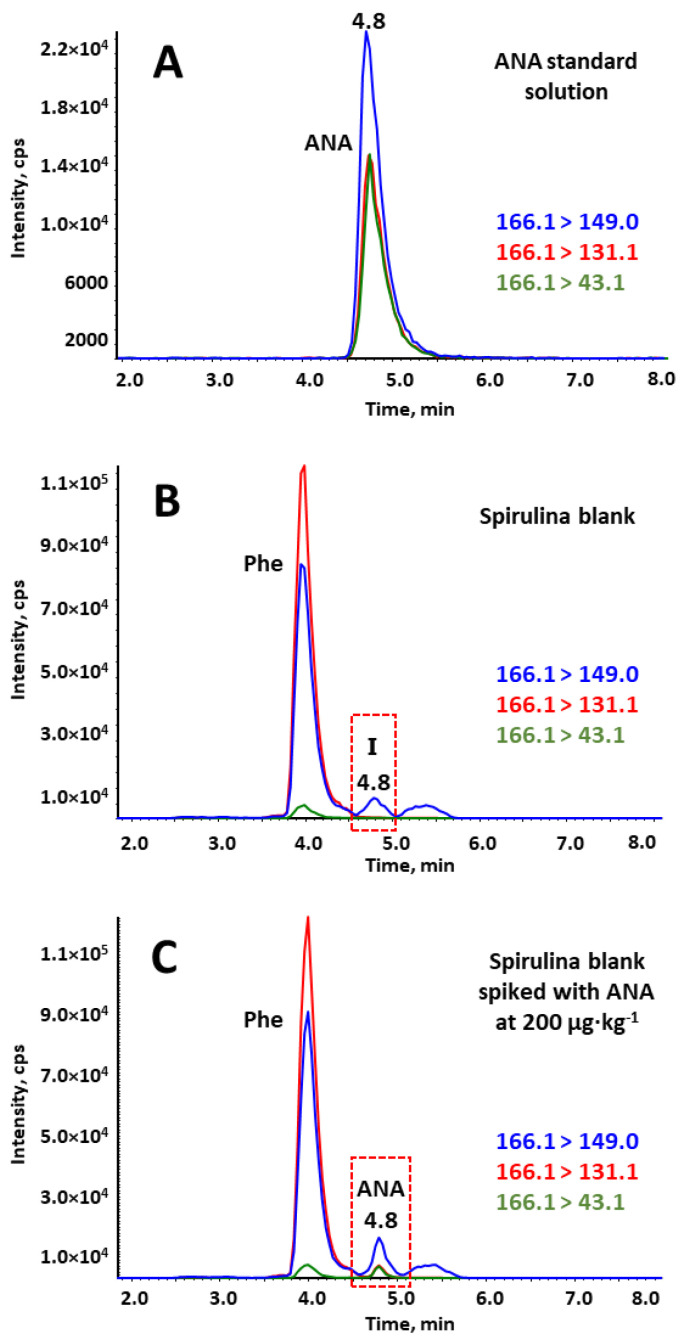
Extracted ion chromatograms of the three most intense transitions of anatoxin-a (ANA) (blue 166.1 > 149.0, red 166.1 > 131.1 and green 166.1 > 43.1). (**A**) Standard solution of ANA at 100 µg·kg^−1^; (**B**) *Spirulina* blank sample; (**C**) *Spirulina* blank sample spiked with ANA at 200 µg·kg^−1^. Phe: Phenylalanine; I: interference.

**Figure 2 toxins-15-00127-f002:**
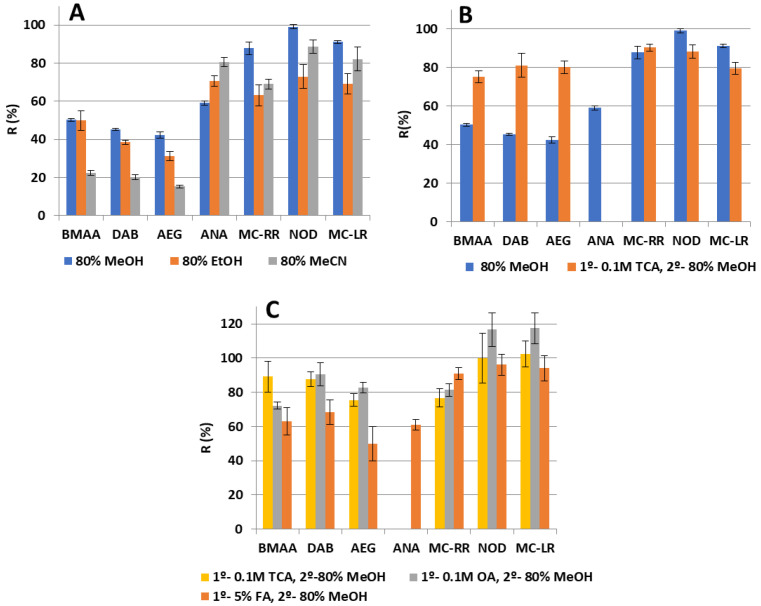
(**A**) Nature of the extraction solvent in the solid–liquid extraction (SLE) performance; (**B**) Performance of SLE extraction when using aqueous 80% methanol (MeOH) extraction versus when adding a previous extraction step with 0.1 M trichloroacetic acid (TCA); (**C**) Performance of different acidic extractions in the first step of the SLE procedure. Error bars represent the standard error.

**Figure 3 toxins-15-00127-f003:**
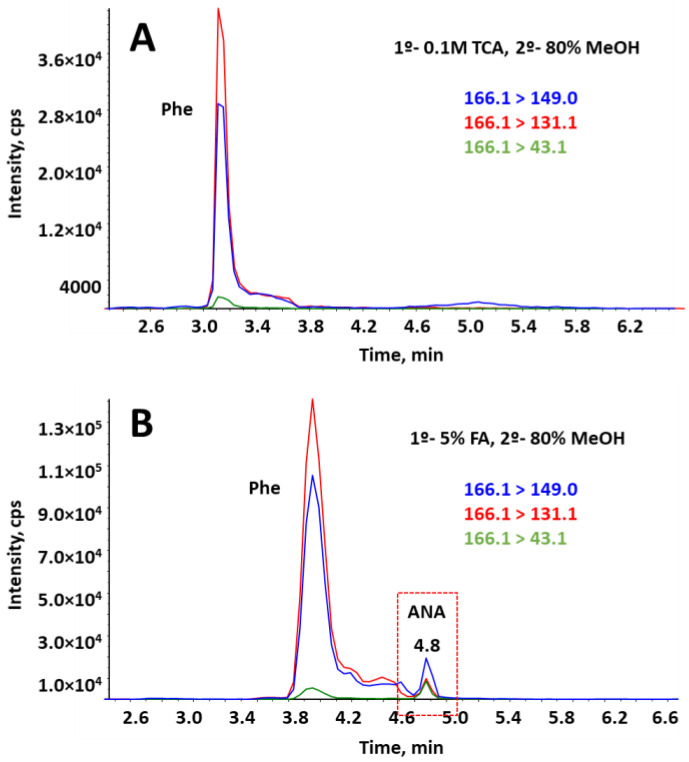
Extracted ion chromatograms of the three most intense transitions of ANA when the SLE is done with (**A**) 0.1 M TCA followed by 80% MeOH; (**B**) aqueous 5% formic acid (FA) followed by 80% MeOH.

**Figure 4 toxins-15-00127-f004:**
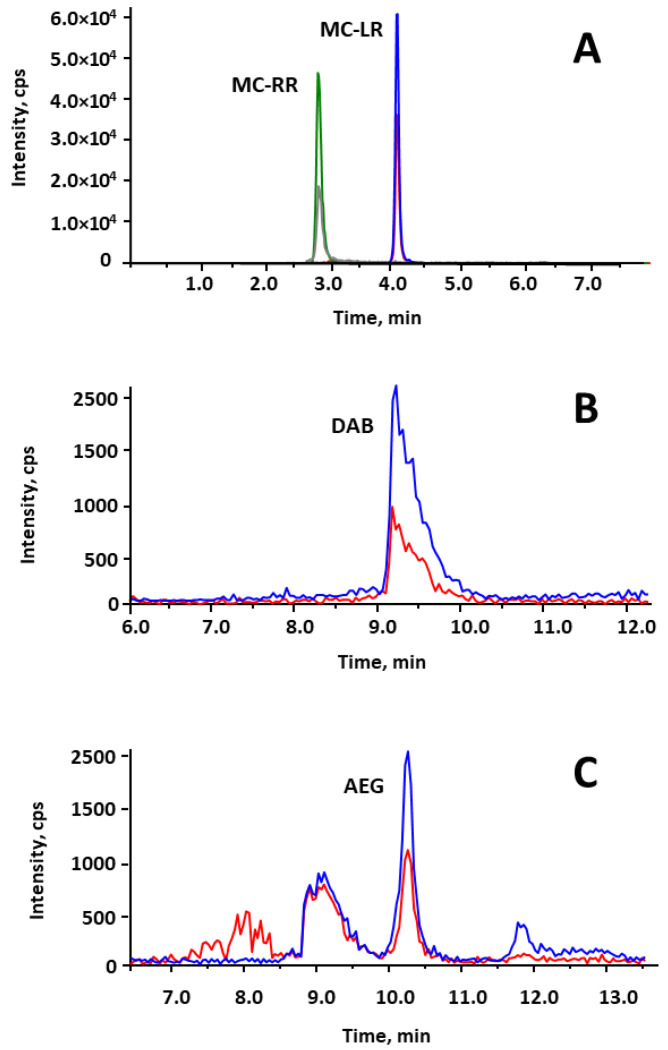
Extract ion chromatogram of different BGA dietary supplement samples contaminated with (**A**) MC-LR and MC-RR, (**B**) DAB and (**C**) AEG.

**Table 1 toxins-15-00127-t001:** Performance characteristics, precision and matrix effect for the proposed solid–liquid extraction–tandem–solid–phase extraction–hydrophilic interaction liquid chromatography–tandem mass spectrometry (SLE-tandem-SPE-HILIC-MS/MS) method in *spirulina* sample.

Analyte	Linear Range (μg·kg^−1^)	LOD (μg·kg^−1^)	LOQ (μg·kg^−1^)	R^2^	Intra-Day Precision (RSD %) n = 9			Inter-Day Precision (RSD %) n = 15			Matrix Effect (%) n = 9		
					L1	L2	L3	L1	L2	L3	L1	L2	L3
MC-LR	60–500	18	60	0.9906	8.3	11.5	11.4	14.9	14.7	14.8	467.8	401.2	356.4
NOD	50–500	15	50	0.9872	11.9	13.1	16.3	15.2	15.3	13.9	138.2	118.7	123.3
MC-RR	50–500	15	50	0.9910	8.8	7.0	3.7	8.2	10.1	5.6	41.0	43.7	38.7
ANA	150–750	45	150	0.9900	9.1	8.0	7.1	8.5	8.6	8.2	33.0	27.6	26.0
BMAA	300–2500	90	300	0.9817	15.2	19.5	19.2	25.1	19.6	18.8	23.0	18.1	9.1
DAB	300–2500	90	300	0.9938	9.8	15.4	9.2	17.3	16.3	11.9	24.4	29.7	21.5
AEG	150–2500	45	150	0.9940	18.3	9.5	15.2	16.3	16.1	19.3	88.0	86.0	63.8

Concentration levels (L1, L2 and L3) for the precision and matrix effect studies were established as follows: 60, 150 and 350 µg·kg^−1^ for MC-LR; 50, 150 and 350 µg·kg^−1^ for NOD and MC-RR; 150, 300 and 600 µg·kg^−1^ for ANA; 300, 800 and 1800 µg·kg^−1^ for BMAA and DAB; and 150, 600 and 1800 µg·kg^−1^ for AEG.

**Table 2 toxins-15-00127-t002:** Recovery assessment for blue-green algae (BGA) dietary supplement samples containing *spirulina*.

Recoveries (%) (n = 6)						
	L1		L2		L3	
Analyte	R %	RSD %	R %	RSD %	R %	RSD %
MC-LR	74.0	15.1	76.6	9.9	81.6	9.2
NOD	102.9	16.1	94.1	19.2	77.7	15.4
MC-RR	91.2	11.6	89.9	5.1	88.0	4.4
ANA	87.8	11.8	80.1	10.2	78.7	9.0
BMAA	70.2	17.5	64.2	16.8	80.7	16.5
DAB	91.0	11.8	76.6	12.5	72.9	11.7
AEG	82.7	11.1	67.9	18.2	81.6	14.0

Concentration levels (L1, L2 and L3) were established as follows: 60, 150 and 350 µg·kg^−1^ for MC-LR; 50, 150 and 350 µg·kg^−1^ for NOD and MC-RR; 150, 300 and 600 µg·kg^−1^ for ANA; 300, 800 and 1800 µg·kg^−1^ for BMAA and DAB; and 150, 600 and 1800 µg·kg^−1^ for AEG.

**Table 3 toxins-15-00127-t003:** Determination of cyanotoxins in BGA dietary supplements (description shown in Supplementary Materials, Appendix A).

Sample	MC-LR	NOD	MC-RR	ANA	BMAA	DAB	AEG
1	n.d.	n.d.	n.d.	n.d.	n.d.	432	n.d.
2	n.d.	n.d.	n.d.	n.d.	n.d.	1065	n.d.
3	n.d.	n.d.	n.d.	n.d.	n.d.	2408	194
4	n.d.	n.d.	n.d.	n.d.	n.d.	900	n.d.
5	n.d.	n.d.	n.d.	n.d.	n.d.	605	n.d.
6	n.d.	n.d.	n.d.	n.d.	n.d.	2038	71
7	n.d.	n.d.	n.d.	n.d.	n.d.	n.d.	n.d.
8	n.d.	n.d.	n.d.	n.d.	n.d.	331	n.d.
9	6583	n.d.	5508	n.d.	n.d.	n.d.	n.d.

Concentration expressed in µg·kg^−1^.

**Table 4 toxins-15-00127-t004:** Multiple reaction monitoring (MRM) parameters for the analysis of cyanotoxins by hydrophilic interaction liquid chromatography coupled with electrospray ionization tandem mass spectrometry (HILIC-ESI-MS/MS).

Analyte	Retention Time (min)	Molecular Ion	Precursor Ion (*m/z*)	DP (V)	EP (V)	CEP (V)	Product Ions (*m/z*)	CE (V)	CXP (V)
MC-LR	2.8	[M + H]^+^	995.6	136.0	10.5	32.0	Q_ion_ 135.2	93.0	4.0
				136.0	10.5	32.0	I_ion_ 105.0	129.0	4.0
NOD	3.0	[M + H]^+^	825.4	96.0	6.5	24.0	Q_ion_ 135.1	89.0	2.0
				96.0	6.5	100.0	I_ion_ 103.2	129.0	4.0
MC-RR	4.0	[M + 2H]^2+^	519.8	41.0	6.0	18.0	Q_ion_ 135.2	39.0	4.0
				41.0	6.0	18.0	I_ion_ 103.2	91.0	6.0
ANA	4.8	[M + H]^+^	166.2	36.0	4.0	10.0	Q_ion_ 131.1	21.0	4.0
				36.0	4.0	10.0	I_ion_ 43.1	39.0	4.0
BMAA	8.9	[M + H]^+^	119.2	26.0	4.0	6.0	Q_ion_ 44.1	27.0	2.0
				26.0	4.0	6.0	I_ion_ 102.1	15.0	4.0
DAB	9.7	[M + H]^+^	119.1	21.0	3.0	14.0	Q_ion_ 101.0	11.0	4.0
				21.0	3.0	14.0	I_ion_ 56.0	31.0	2.0
AEG	11.3	[M + H]^+^	119.1	26.0	3.5	10.0	Q_ion_ 102.0	11.0	4.0
				26.0	3.5	10.0	I_ion_ 56.0	25.0	4.0

Abbreviations: declustering potential, DP; entrance potential, EP; collision entrance potential, CEP; collision energy, CE; collision cell exit potential, CXP; quantification ion, Q_ion_; identification ion, I_ion_.

## Data Availability

Not applicable.

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
