# Peer review of "Determination of Multiclass Cyanotoxins in Blue-Green Algae (BGA) Dietary Supplements Using Hydrophilic Interaction Liquid Chromatography-Tandem Mass Spectrometry"

_toxins, 2023, doi:10.3390/toxins15020127_

Round 1

Reviewer 1 Report

This manuscript establishes an analytical method for the analysis of multiclass cyanotoxins in blue-green algae dietary supplements and optimizes the method. This is an interesting piece of work to achieve simultaneous extraction and analysis of hydrophilic and lipophilic cyanotoxins. The topic is relevant and significant, and the work is well written and is easy to understand. However, a number of improvements are required before acceptance. My specific recommendations for each section are given below.

Introduction

*The introduction should be written in a logical and focused manner. Dividing the introduction into paragraphs can make the presentation more hierarchical and logical. The entire manuscript is based on SLE and SPE techniques, so more attention should be paid to their strengths and weaknesses. Some sentences are too lengthy and suggested to be reduced.

*Lines 38-45: These lines seem out of place, compared to before and after - maybe they could be moved to the end or a more appropriate place of the introduction.

*Line 63: Since saxitoxins and cylindrospermopsin are mentioned, which are also common toxins, why does the method not include them?

Results

*The total ion chromatograms or extracted ion chromatograms of the spiked sample should be supplemented to display chromatographic information such as separation, retention time and peak shape of each analyte.

*Lines 150-161: Nice to include an explanation of ANA identification, which will help to avoid false positives. 

*Line 201: Can the authors clarify why 80:20 was chosen for the three organic solvents?

*Line 281: A decrease in recoveries with increasing sample amount does not necessarily mean a decrease in the response signal of the analytes.

*Line 284: Nice to include comprehensive methodological validation results here. However, the comparison between this result and other existing results should be strengthened to highlight the advantages of this proposed method.

*Line 388: Four toxins were detected from the BGA supplements, but only two were shown in Figure 4. I hope the authors can also provide EICs of DAB and AEG.

Methods

*Line 469: ZIC-HILIC column was used for simultaneous analysis of all analytes. Can the authors explain whether the hydrophilic interaction column can effectively retain lipophilic MCs and NOD? If retention is weak, is the analyte easily co-eluted with other interferents in the actual sample testing? Do the co-elution compounds have an effect on the target analyte?

Reviewer 2 Report

The authors present an interesting and useful method for the determination of multiple cyanotoxins present in health food supplements originating from cyanobacterial sources. The idea of a multi-analyte method for cyanotoxins from different families is not especially novel, but it is good to see further information presented which facilitates the analysis of the most important toxins.

In terms of determinant choice, I approved of the section of analyte classes, specifically incorporating microcystins, nodularin and ATX. The inclusion of BMAA is interesting also, and again useful, but really it would have been appropriate to include all relevant BMAA-related analogues which are typically assessed as part of any BMAA analysis. For me the absence of BAMA is a missed opportunity. In terms of MC analogues, I was a bit disappointed to see only the inclusion of two MC analogues (MC-LR and MC-RR). This is also a missed opportunity as any laboratory conducting MC analysis will need/want to include a wider range of MCs – so an important step for any lab taking this method on would be the refinement to incorporate other MC forms. I would also ideally have incorporated saxitoxin, being another hydrophilic cyanotoxin potentially present within BGA samples. This is not to say that the work is irrelevant, but as it stands, the work is more of a proof of concept for a multiclass method, with the next step incorporating addition of other important toxins being essential before the method would be useful for routine testing and any potential interlaboratory assessment (which would be recommended).

The introduction contains a good overview of the background, including some important information and references – good also that the three genera were specifically discussed. There is a great deal of thorough investigation that is a valuable resource to the reader, so congratulations on this. I would say that some work is required on making the discussions clearer to follow. Use of more paragraphs to separate thematic content is one thing. Another is being absolutely clear on exactly what is conducted and what experimental parameters were utilised. Specific comments on this are given below. Finally, I strongly recommend that more samples are assessed using this method. Given the method is straight forward and high throughput, then please subject the method to more samples, and use this data to importantly describe occurrence in a more statistically-relevant number of samples.

·         Abstract: L9 – please change to “including the microcystins MC-LR and MC-RR,” (as currently the wording infers that other MC analogues are also incorporated)

·         L14 – specific the number of supplements tested – currently “different supplements” is not specific enough information for the reader

Introduction

·         L32 – please quantify “an important proportion” with appropriate reference(s)

·         L34-35 – please provide reference for evidencing statement regarding impact of environmental factors on cyanobacterial growth

·         Page 2 onwards, please use separate paragraphs to separate clearly into subject-related themes. There are at least 3-4 different subjects in the introduction which would be clearer. Especially make it clear where the paper’s targets and aims start.

·         L80 and around. As I recall, the State of Oregon introduced a health-related regulation for microcystins in food supplements (1mg/kg MC in BGA-containing products). This is referenced in Gilroy et al 2000 Environ Health Perspect 108:435–439 (but please check if more recent references exist) as this additional angle is relevant to include: https://www.oregon.gov/oha/ph/HealthyEnvironments/Recreation/HarmfulAlgaeBlooms/Documents/Gilroy_Algae_Supp.pdf

·         L96 – please reword “without previous derivatization” to something like “without pre-analysis derivatization” – as currently not clear what is meant

·         L95 – separate long sentence into two sentences

·         L130 please describe why these analytes were chosen in preference to other important compounds

Results

·         L208 – please provide information and references for the tandem SPE method

·         Section 2.2. The work conducted here is great, and the information really valuable to describe. However, currently this reads more like a laboratory notebook, rather than a published paper. Please can you reword this section, with use of sub-sections, to more clearly define the various investigations conducted and the ultimate outcomes, just to improve the clarity and ease of reading

·         L302-304 – please describe why these concentration ranges were utilised. Particularly noting the Oregon State Health lab regulatory limit, which is higher than the top level used here for MCs

·         Table 1. LOQ concentrations seem very high in comparison to various other published validation manuscripts. Please describe why this is the case and justify how these are fit for purpose

·         Matrix effects – the data generated shows worrying levels of matrix effects, especially with 400% enhancement for MC-LR and the significant suppression for BMAAs and AZA (BMAA is <10% of expected so huge suppression). Really MC-RR is the only acceptable result in this case. This is an important finding that needs greater justification than currently provided. Specifically there is an important need to describe that exactly is meant by a procedural calibration curve and how this resolves any matrix-affected method accuracy issues.

·         L360 – please describe what constitutes “acceptable precision” and how this is assessed

·         2.3.4 recovery determination. Please clarify exactly what was conducted to determine recovery and exactly what recovery is being assessed. The authors here compared peak areas before and after “sample treatment”. But it is not clear what sample treatment is being referred to. The authors need to be totally clear as to whether this is a full method recovery assessment, or extraction recovery or extraction + SPE only recovery. Ideally, the recoveries should be full method recovery experiments, with recovery determined with full quantitation of analyte concentrations in spiked samples subjected to the full method from start to finish. Comparison of peak areas does not go far enough to properly establish method recovery.

·         2.4 – nine samples were analysed in total if I understood correctly. In my opinion this does not really constitute a particularly thorough assessment of toxin concentrations in what is a very wide range of products on the commercial market. In addition to the number not being sufficient, the information regarding the BGA genus content should be summarised and discussed in the main body of text – not just Table S3. This is critical, as without reference to supplementary materials, there is no information obtainable regarding the proportions of BGA products which contain toxins, for each genus type. My recommendation here is for the authors to analyse more samples (which are easily obtainable commercially) and to subject these to testing using their routine method, and to build up a larger data set – then discuss these findings accordingly

If these issues can be addressed I believe this manuscript would be worthy of publication in Toxins.

Reviewer 3 Report

Major Comments:

The major goals of the study appear to be testing extraction and separation parameters for detection of a range of cyanobacterial toxins. Extraction solvents with and without various acids were tested and extraction of each toxin evaluated by HILIC HPLC separation with tandem mass spectrometry (MS/MS) detection. The authors concluded that different combinations of extraction solvent impacted anatoxin most dramatically. The optimized extraction solvent was tested for repeatability and precision using spiked standards prior to applying the method to blue-green algal (BGA) nutritional supplements.  

The authors described the methods used in appropriate detail and described the results of the method testing with appropriate statistical rigor. The analysis of the BGA samples left something to be desired in the presentation of the data. Text and one figure were dedicated to the results for these experiments, however a table with the associated measures of confidence in the results for each of the nine samples would have been appropriate. Less statistical rigor was used to describe the results of those experiments, such as precision and reproducibility in repeated extracts of the real world samples would have been appropriate given that this application was highlighted as the purpose of the paper in the title.

Minor Comments:

Line 405-406: “This does not entail that MC-LR is the only toxic or that the rest of cyanotoxins are less harmful. It merely reflects the lack of toxicological information on these compounds.”

This sentence introduces confusion. Do the authors mean to say that MC-LR was the only toxin detected above a recommended TDI? It is unclear if the authors also mean to say that the “toxicological information” (i.e. meaning the toxicity in LD50 values?) is only available for that members of the microcystin family? Is it that the TDI has not been established for all compounds on your target list? It appears that you based a 1 ppm (mass/mass) quantity threshold for all based on the MC-LR number, but  please clarify.

Round 2

Reviewer 2 Report

Thanks to the authors for taking the time to respond to the comments made and suggestions. I can see that appropriate changes were made, and justifications provided where experiments were not changed following suggestions.